# Estimation of immune cell content in tumour tissue using single-cell RNA-seq data

Max Schelker[1,2], Sonia Feau[1], Jinyan Du[1], Nav Ranu[1], Edda Klipp [2], Gavin MacBeath[1], Birgit Schoeberl[1] & Andreas Raue [1]

As interactions between the immune system and tumour cells are governed by a complex network of cell–cell interactions, knowing the specific immune cell composition of a solid tumour may be essential to predict a patient's response to immunotherapy. Here, we analyse in depth how to derive the cellular composition of a solid tumour from bulk gene expression data by mathematical deconvolution, using indication-specific and cell type-specific reference gene expression profiles (RGEPs) from tumour-derived single-cell RNA sequencing data. We demonstrate that tumour-derived RGEPs are essential for the successful deconvolution and that RGEPs from peripheral blood are insufficient. We distinguish nine major cell types, as well as three T cell subtypes. Using the tumour-derived RGEPs, we can estimate the content of many tumours associated immune and stromal cell types, their therapeutically relevant ratios, as well as an improved gene expression profile of the malignant cells.

[1] Merrimack Pharmaceuticals, Inc., Cambridge, MA 02139, USA. [2] Humboldt-Universität zu Berlin, Unter den Linden 6, 10099 Berlin, Germany. Correspondence and requests for materials should be addressed to A.R. (email: araue@merrimack.com)

Enhancing a patient's immune response to cancer using immune checkpoint inhibitors is arguably the most exciting advance in the treatment of cancer in the past decade[1,2]. Unfortunately, only a subset of patients (typically ~20%) show long-lasting responses post checkpoint blockade[3]. Combining prospective patient selection based on predictive response biomarkers (=precision medicine) and immunotherapy has the potential to further transform patient care. To date, it has been shown that location and abundance of immune cells are prognostic for predicting patient outcome on standard therapy[4,5]. In addition, for checkpoint inhibitors-like anti-PD1, anti-PDL1, and anti-CTLA4 agents, the presence of relevant T cell populations correlates with treatment efficacy[6]. Thus, it is likely that the key to predicting response to immunotherapy lies in the patient-specific immune cell composition at the site of the tumour lesion.

In theory, it is possible to infer the immune, tumour, and stroma cell content of a solid tumour from its bulk gene expression profile if reference gene expression profiles (RGEPs) can be established for each tumour-associated cell type. Mathematically, this class of inverse problems is known as *deconvolution*[7]. To date, deconvolution of bulk gene expression has been described and validated for haematological malignancies[8,9], where RGEPs can be established from peripheral blood mononuclear cells (PBMCs). This approach has been applied theoretically to solid tumours[10], but until recently it has been impossible to validate this extrapolation experimentally. It has been difficult to obtain RGEPs for cell types that are not available in the peripheral blood, such as endothelial cells (ECs) and cancer-associated fibroblasts (CAFs) and it remains unclear to which extent the gene expression profile of an immune cell changes upon tumour infiltration. With the advent of the single-cell RNA sequencing (scRNA-seq) technology, however, it is now possible to determine gene expression profiles for tumour-infiltrating immune cells, tumour-associated non-malignant cells, and individual tumour cells from the same solid tumour biopsy.

We collected and investigated RNA-seq gene expression profiles of more than 11,000 single cells from three distinct primary human tissue sources: To characterise cells associated with the tumour microenvironment we accessed data from 19 melanoma patients[11], to characterise the baseline immune cell gene expression we accessed data from PBMCs originating from four healthy subjects[12] and last, we generated immune and tumour cell gene expression profiles from four ovarian cancer ascites samples in-house. In the following, we show that gene expression profiles from tumour-associated immune cells and from PBMCs differ substantially. Therefore, reference profiles obtained from PBMCs are insufficient to deconvolve the bulk profile of a melanoma tumour sample. We find that indication-specific immune cell RNA-seq profiles from different patients are sufficiently similar to each other to define a *consensus* profile for each cell type, and that these consensus profiles enable accurate deconvolution of bulk tumour profiles. Our results show that the generation of specific RGEPs is both necessary and sufficient to enable reliable estimation of tumour composition from bulk gene expression data. Our approach resolves tumour-associated cell types that cannot be estimated by RGEPs derived from PBMCs. We can identify nine different cell types including immune cells, CAFs, ECs, ovarian carcinoma cells and melanoma cells. In addition, RGEPs for immune cells can be used to estimate the unknown gene expression profiles of tumour cells from bulk gene expression data patient specifically. Our work emphasises the importance of generating RGEPs specific to each indication of interest.

## Results

**Gene expression of cells in the tumour microenvironment.** First, to investigate the extent to which gene expression profiles change as immune cells move from peripheral blood to the tumour microenvironment, we compared immune cell scRNA-seq profiles across three human data-sets: (1) data-set of 4000 single cells derived from peripheral blood of four healthy subjects[12]; (2) data-set of 4645 tumour-derived single cells from 19 melanoma patient samples[11] and an unpublished data-set of 3114 single cells from four ovarian cancer ascites samples. Single-cell RNA-seq data requires careful data processing and normalisation particularly when comparing data originating from different sources and sequencing technologies. To characterise the single cells and to illustrate genome wide similarities and differences in their gene expression profiles, we applied the dimensionality reduction technique t-distributed stochastic neighbour embedding (t-SNE)[13]. This is an unsupervised machine learning algorithm that places each single cell into a two-dimensional plane. Cells with gene expression profiles that are similar are placed close to each other and farther apart if they are more different. Figure 1a shows that clusters associated with specific cell types and from different data sources emerge spontaneously. The t-SNE map with data source-specific colour coding is shown in Supplementary Fig. 1 to visualise the cell-specific rather than data source-specific clustering. Using the aggregated single-cell data-set, we developed a classification approach that can identify cell types irrespective of the data source. We can identify and classify nine major cell types: T cells, B cells, macrophages/monocytes, natural killer (NK) cells, dendritic cells (DCs), CAFs, ECs, ovarian carcinoma cells, and melanoma cells. All remaining cells that fail to pass the classification threshold for any specific cell type are assigned as "unknown". Interestingly, the "unknown" cells are mostly located in the T cell clusters, suggesting that some T cells are more difficult to classify than cells from other cell types. However, the percentage of "unknown" cells per sample is generally very low (<0.03%). Further, we could classify T cells into three subtypes: CD4+, CD8+, and regulatory T cells (Treg). The ratios of CD4+ or CD8+ T cells and immune suppressive Tregs were suggested as markers for immunologically active vs. inactive tumours[6]. Although our methods can easily be extended to include additional cells and further subdivisions, we limited ourselves to the nine major cell types to benchmark our classification algorithm. As previously reported[11] and shown in Supplementary Fig. 2, malignant tumour cells and associated fibroblasts cluster by patient and non-malignant cells cluster by cell type. Tumour biopsies should contain immune cells from tumour blood vessels and from recently extravasated immune cells. Therefore, a partial overlap between PBMC and tumour-associated immune cells is expected. We analysed pair-wise similarities between the averaged gene expression profiles of each identified cluster. This analysis is more quantitative and robust to noise as the single cell comparisons. The results shown in Fig. 1b indicate that most clusters, while distinct, are most closely associated with clusters from the same cell types. This is an important quality control step that confirms that potential batch effects are successfully alleviated by the data processing and normalisation strategy (see "Methods" section). Tregs seem to be most distinct across the three different data-sets potentially indicating different context-dependent subsets[14]. However, the microenvironment has a clear and quantifiable impact on gene expression. In the following, we will address the question if gene expression profiles based on PBMCs are good approximations for what is observed in the tumour microenvironment and how the PBMC-derived gene expression profiles impact the quality of deconvolution of bulk expression data.

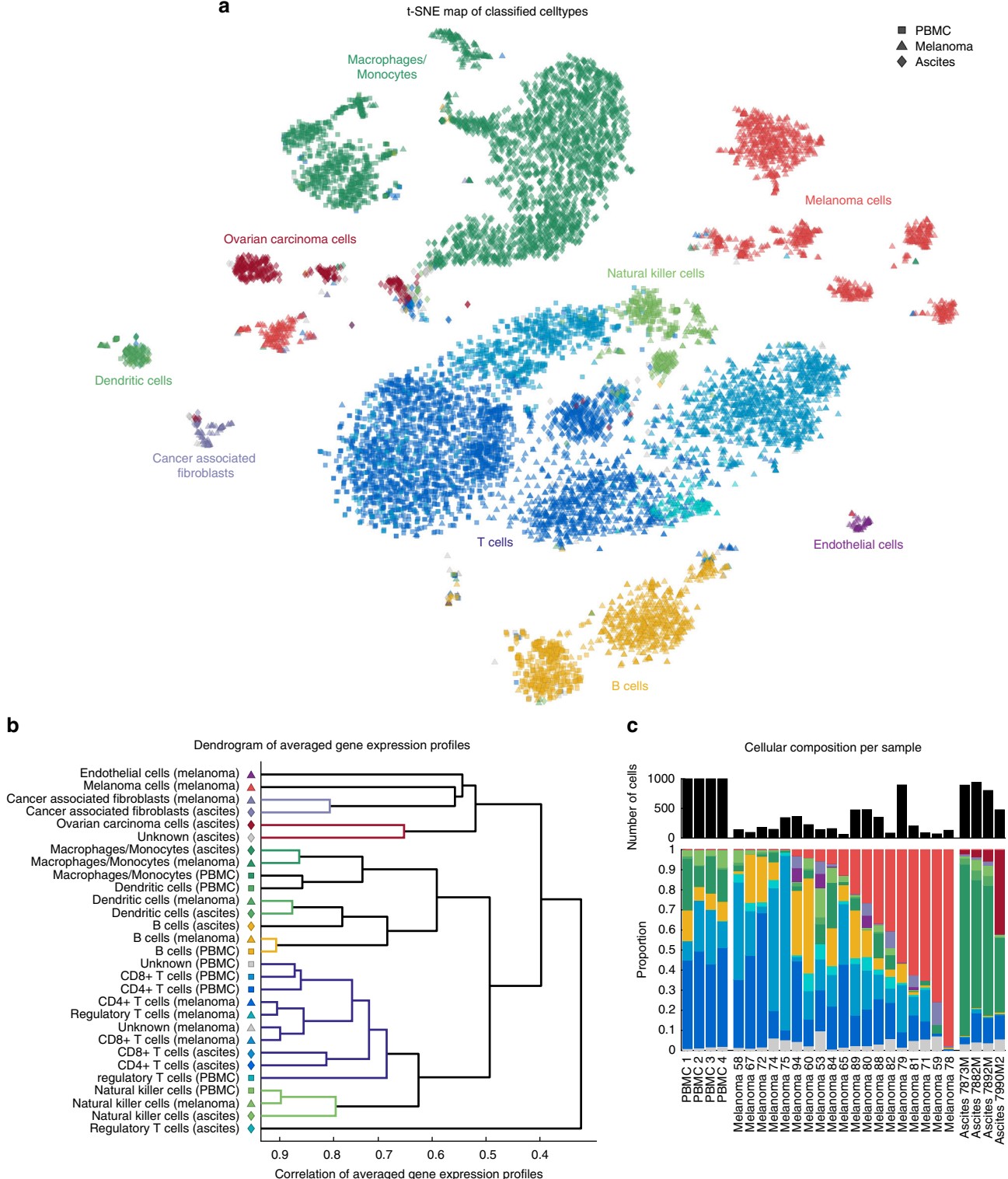

**Fig. 1** Comparison of gene expression profiles of single cells from different data sources. **a** Single cells were arranged in two dimensions based on similarity of their gene expression profiles by the dimensionality reduction technique t-SNE. The clusters that emerge spontaneously can be associated with cell types (colours) and data source (symbol types: squares for PBMC-data sets, triangles for the melanoma-data sets, and diamonds for ascites data-sets). **b** Pair-wise correlation of averaged gene expression profiles of clusters encoding cell type and origin as identified in **a** visualised as dendrogram. **c** Number of cells and cellular composition per sample

First, we observed that the frequency of each cell type appears to be distinct for each sample as depicted in Fig. 1c. The cellular composition of the PBMC samples from different donors is more similar to each other compared to the cellular composition across the ascites or melanoma samples. We validated the predicted cellular composition based on our scRNA-seq-based classification with previously reported results for all melanoma samples[11]. Also, we compared the predicted cellular composition for all ascites

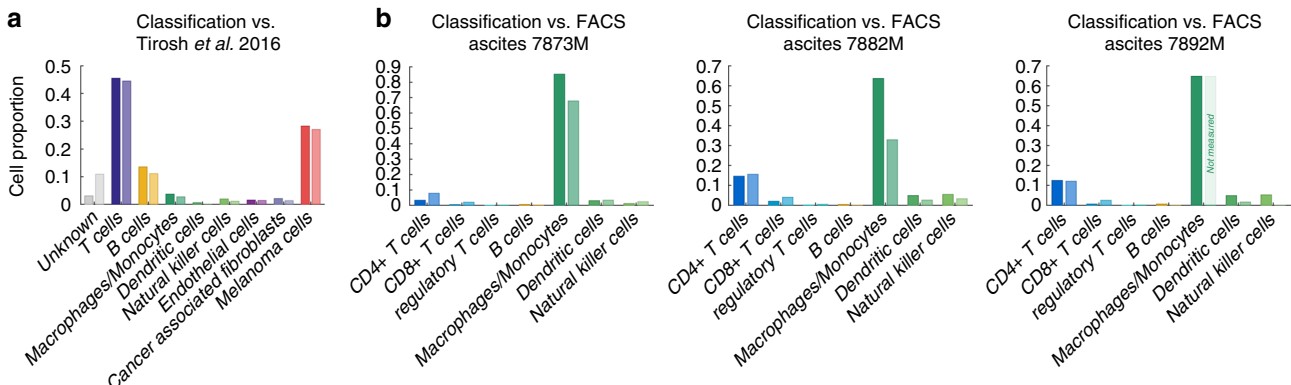

**Fig. 2** Benchmarking the cell type classification to literature and experimental FACS analysis. **a** The result of our cell type classification (left bars, dark colours) compared to the cell types provided across all melanoma samples in the data-set by Tirosh et al. [11,22] (right bars, light colours). **b** Cell type classification (left bars, dark colours) compared to FACS data (right bars, light colours) for three ovarian ascites patient samples. For sample 7892M, macrophages/monocytes quantification is missing for FACS

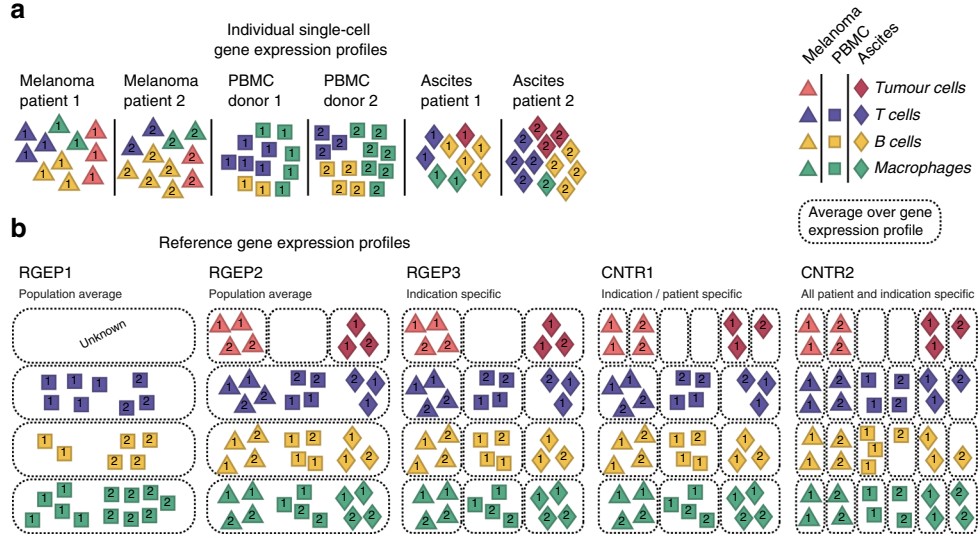

**Fig. 3** Construction of five RGEPs for benchmarking the estimation accuracy. **a** For each source location (melanoma, ascites, PBMC) individual single cell gene expression profiles are collected for multiple patients. Colours indicate the cell type, numbers indicate the patient sample and symbols show the source location (triangles for melanoma, squares for PBMCs, and diamonds for ascites). **b** Construction of REGPs from three single cell data-sets: RGEP1 bases on the population average of PBMC data; RGEP2 takes all three source locations into account; RGEP3 is indication-specific and location-specific; CNTR1 is patient-specific for tumour cells and indication/location-specific for non-malignant cells; CNTR2 is fully patient-specific

samples experimentally with fluorescence-activated cell sorting (FACS). As depicted in Fig. 2 our classification is in line with previously published results and our FACS measurements.

**Using single-cell data as a benchmark for deconvolution.** The microenvironment-specific gene expression profiles of immune cells, as well as the true composition of a given sample can be obtained by scRNA-seq and can serve as ground truth to benchmark deconvolution approaches. We studied how the deconvolution results of bulk gene expression data, for instance of a melanoma sample, are affected by microenvironment-specific changes and by patient-to-patient variation. As benchmark for the deconvolution, we constructed artificial "bulk" gene expression data by aggregating all single-cell gene expression data for each of the 27 samples, as well as different sets of REGPs by different strategies for averaging over tissue sources and patients. We compare the inferred, a priori known cellular composition of a given sample using five different RGEPs (see Fig. 3 for illustration): The first, RGEP1 is derived from the PBMC data-set

only. Therefore, estimates for tumour-associated cell types will not be available in this case. The second, RGEP2, is derived for each cell type across the three data-sets (PBMC, melanoma, and ascites). The third, RGEP3 is data-set/indication-type and cell-type specific. As additional benchmarks, we set up two control scenarios (CNTR1 and CNTR2) that are extensions of RGEP3 and include patient-specific information. These scenarios are, of course, not applicable in the real world, but serve to evaluate the relative importance of patient-specific information. CNTR1 uses patient-specific profiles for the malignant cells only and consensus profiles for each non-malignant cell type. CNTR2 uses patient-specific profiles for all cell types. In principle, CNTR2 serves as the upper limit on what is technically possible using deconvolution approaches.

**Origin and quality of RGEP determine deconvolution results.** To compare the five possible RGEPs and their impact on deconvolution accuracy, we estimated the cellular composition from the 27 constructed bulk expression datasets using the

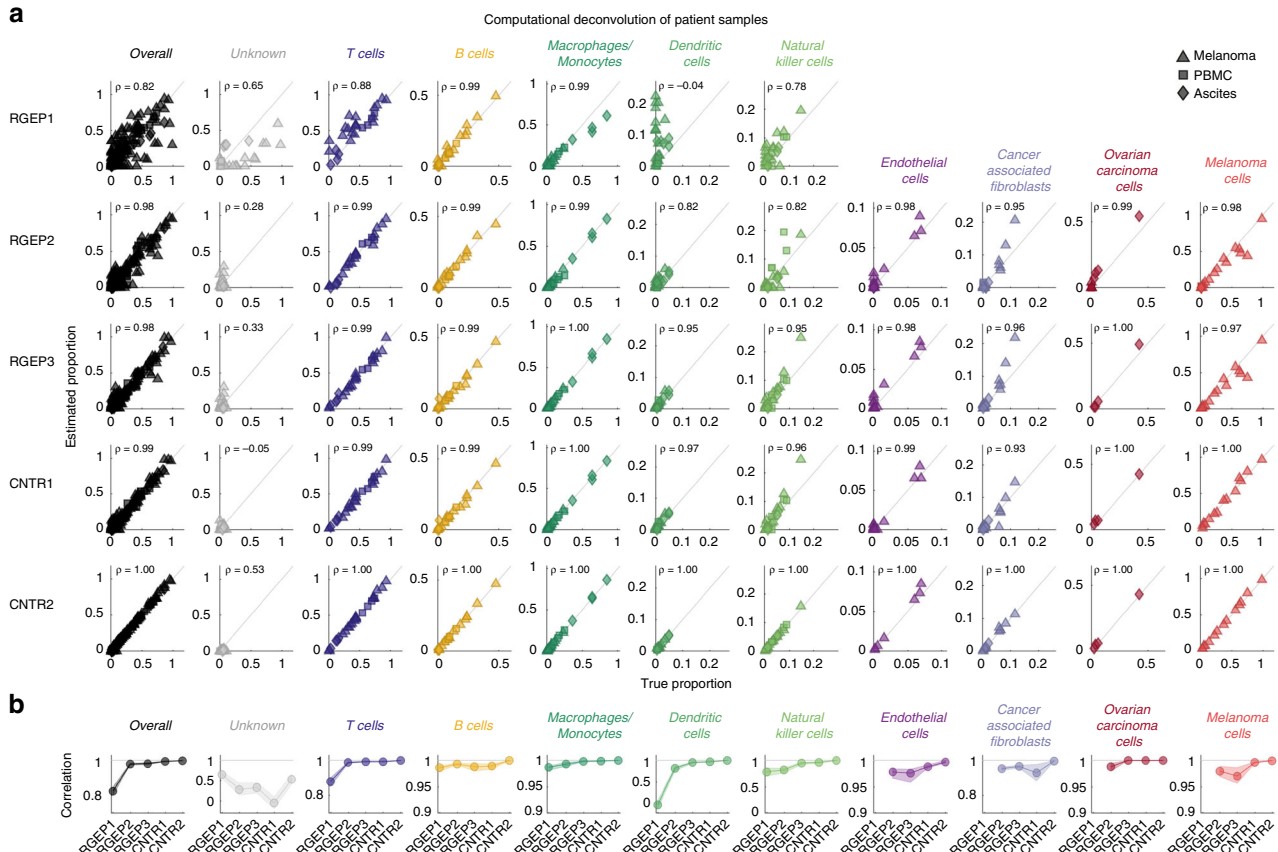

**Fig. 4** Estimation accuracy of cellular composition is dependent on the origin and quality of RGEPs. **a** Scatter plot of true and estimated cell proportions for all 27 patient samples. Each dot represents one patient sample. Values close to the diagonal correspond to high deconvolution accuracy. Columns depict cell types; rows describe the five different configurations (REGP1-3 and CNTR1-2). $\rho$ denotes the Pearson's correlation coefficient. In configuration REGP1, estimates for tumour-associated cell types are not available. **b** Pearson's correlation coefficient between estimated and true cell fraction for all five configurations. Dots denote the median of the correlation coefficient; the shading represents the uncertainty based on bootstrapping (upper and lower quartile). (Please note the different scaling of the figure axes.)

CIBERSORT deconvolution method[8]. This method is designed to be more robust against noise, unknown mixture content and closely related cell types. CIBERSORT has been shown to outperform other methods based on in vitro cell mixture benchmarks. The CIBERSORT algorithm was originally developed for the deconvolution of microarray data. Here, we show that the algorithm can be applied to RNA sequencing data, as well if RGEPs derived from the same technology are used to characterise the cell types. All deconvolutions were performed using a collection of genes, which comprises 1076 signature genes that were found to maximally differentiate various cell types[8,11]. For each cell type, the estimated proportion was compared to the true proportion in the 27 constructed samples (Fig. 4a). The Pearson correlation coefficients between estimated and true cellular composition were used as a measure of prediction accuracy (Fig. 4b). Qualitatively similar results are obtained by using the root-mean-square deviation (RMSD) (see Supplementary Fig. 3). For the T cells, each subset was estimated individually. In Fig. 4, the estimates of all T cell subsets were added to obtain the total T cell proportion for each sample. The results for the individual T cell subsets are considered separately in Fig. 5. Overall, estimations based on RGEP1 were less accurate (Pearson correlation $\rho$ = 0.82) than for RGEP2 and RGEP3 or for CNTR1 and CNTR2 (Pearson correlation $\rho \geq 0.98$). For RGEP1, due to the unavailable reference profiles for tumour-associated cell types the true proportion of unknown cells is larger than for the other RGEPs and the estimation quality is mediocre (Pearson correlation $\rho$ = 0.65).

For RGEP2 and RGEP3, as well as for CNTR1 and CNTR2, the true proportion of unknown cells is negligibly small. Correlation is not a good measure of accuracy in case the true proportion of cells is small. For RGEP1 the estimation performs well for T cells (Pearson correlation $\rho$ = 0.88, not distinguished into subtypes here), B cells (Pearson correlation $\rho$ = 0.99) and macrophages/monocytes (Pearson correlation $\rho$ = 0.99). However, the accuracy improves further for all other settings (Pearson correlation $\rho \geq$ 0.99). For RGEP1 the estimation for DCs (Pearson correlation $\rho$ = −0.04) is poor and mediocre for NK cells (Pearson correlation $\rho$ = 0.78). The estimation for DCs improves considerably for RGEP2 (Pearson correlation $\rho$ = 0.82) and RGEP3 (Pearson correlation $\rho$ = 0.95). The estimation for DCs still improves slightly for CNTR1 (Pearson correlation $\rho$ = 0.97) but reaches its maximum only for CNTR2 (Pearson correlation $\rho$ = 1.00), indicating that gene expression of DCs is heavily dependent on the source of isolation, which is in agreement with the evidence that distinct subsets of DCs are highly specialised in the generation of immunity[15]. The estimation for NK cells improves slightly for RGEP2 (Pearson correlation $\rho$ = 0.82) and reaches close to optimal in RGEP3 (Pearson correlation $\rho$ = 0.95) compared to CNTR1 (Pearson correlation $\rho$ = 0.96) and CNTR2 (Pearson correlation $\rho$ = 1.00). For RGEP2 to CNTR2, estimates for the tumour-associated cell types (CAFs, ECs and the malignant cells) become available and are estimated accurately (Pearson correlation $\rho \geq 0.95$). Interestingly, the estimation for the malignant cells does not improve much upon inclusion of patient-specific

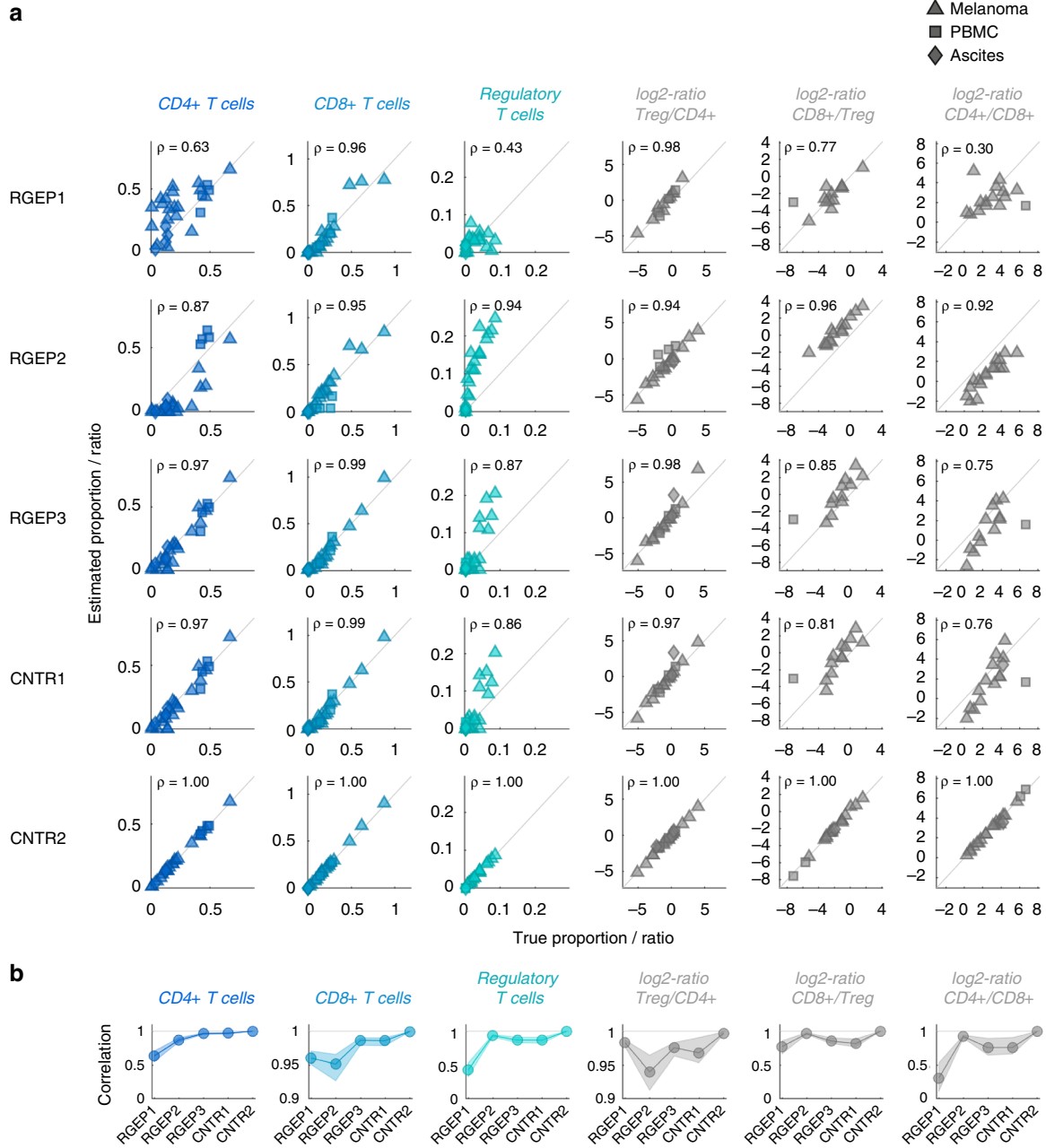

**Fig. 5** Estimation accuracy of T cell subsets within the T cell population and clinically relevant ratios of T cells and their dependence on the origin and quality of RGEPs. **a** Scatter plot of true and estimated cell proportions for all 27 patient samples. Each dot represents one patient sample. Values close to the diagonal correspond to high deconvolution accuracy. Columns depict cell types; rows describe the five different configurations (REGP1-3 and CNTR1-2). $\rho$ denotes the Pearson's correlation coefficient. In configuration REGP1, estimates for tumour-associated cell types are not available. **b** Pearson's correlation coefficient between estimated and true cell fraction for all five configurations. Dots denote the median of the correlation coefficient; the shading represents the uncertainty based on bootstrapping (upper and lower quartile). (Please note the different scaling of the figure axes.)

information, suggesting that deconvolution using consensus profiles is feasible. This is possible because the tumour cells are in general very different from the non-malignant cells which make their deconvolution easier (see Fig. 1b). For CNTR2, ECs and CAFs have an increased accuracy (Pearson correlation $\rho = 1.00$) compared to the other settings (Pearson correlation $\rho \sim 0.95$), indicating that gene expression of those cell types is influenced by patient-specific microenvironment. Interestingly, when considering the distance from the bisection (as shown in Fig. 4) as measure of estimation accuracy, we find that it is independent of the true cell type proportion. However, the overall accuracy is different for each cell type.

Given the importance of T-cell ratios for treatment outcome[6], we further analysed the estimation accuracy for T cell subsets as well as for therapeutically relevant T cell ratios (Fig. 5). Surprisingly, for CD8+ T cells, the estimation results are accurate (Pearson correlation $\rho \sim 0.95$) for all RGEPs. For CD4+ and regulatory T cells, the estimation results using RGEP1 are only mediocre (Pearson correlation $\rho = 0.63$ and $\rho = 0.43$) but improve significantly for RGEP2 (Pearson correlation $\rho = 0.87$ and $\rho = 0.94$). This is also reflected in the ratios of Treg/CD4+, CD8 +/Treg, and CD4+/CD8+ T cells that reach accurate estimations for RGEP2 (Pearson correlation $\rho = 0.94$, $\rho = 0.96$, and $\rho = 0.93$). The estimation for all T cell subsets and ratios does not

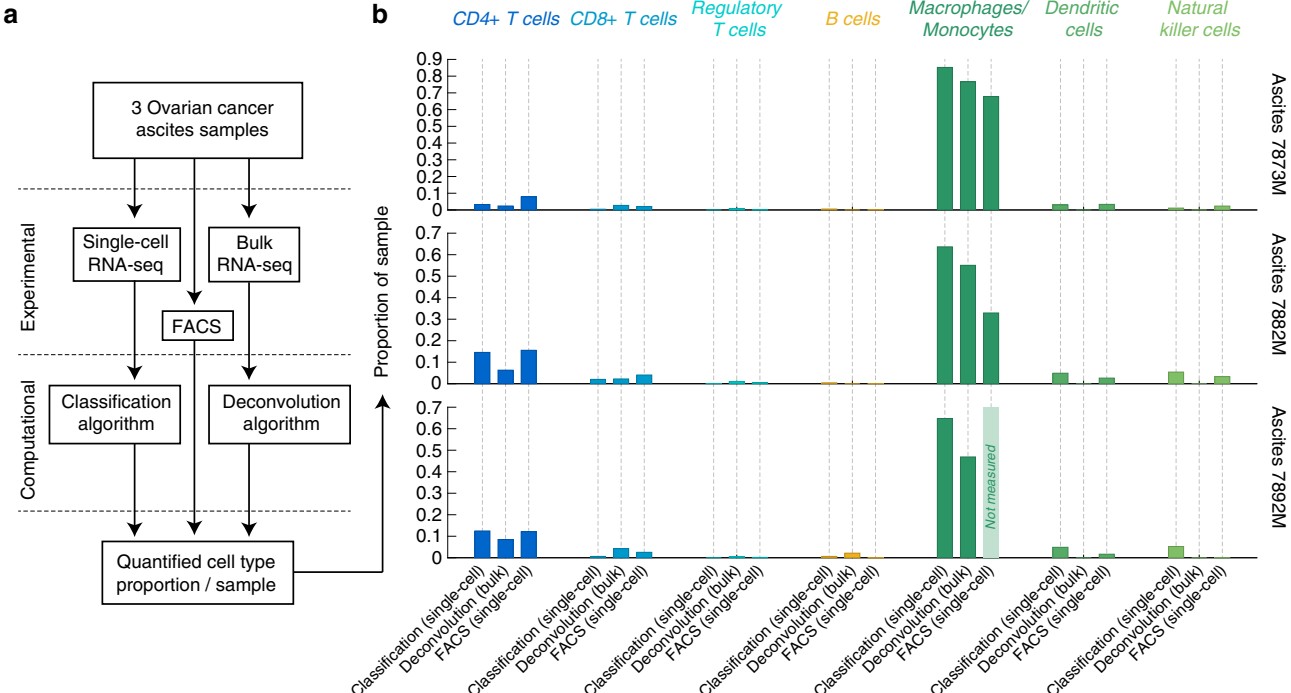

**Fig. 6** Estimates of cellular composition by three different methods. Three ovarian cancer ascites samples were profiled by single-cell and bulk RNA sequencing, as well as FACS. **a** Process diagram of data and results generation for three ovarian cancer ascites samples. **b** Estimates of the cellular composition are derived by: (1) classification based on single-cell RNA sequencing data; (2) computational deconvolution on the bulk RNA sequencing data using the single-cell RNA sequencing derived RGEP3; (3) quantification by FACS. For sample 7892, macrophages/monocytes quantification could not be determined by FACS

significantly improve for CNTR1 but does improve for CNTR2 (Pearson correlation $\rho = 1.00$), indicating that gene expression of T cells is influenced by the patient-specific microenvironment. In summary, deconvolution using consensus gene expression profiles based on indication-specific gene expression profiles (RGEP3) were sufficient to obtain reliable estimates of the cellular composition of the samples without requiring patient-specific data on the individual cell types. Deconvolution using gene expression profiles based on data from peripheral blood (RGEP1) or based on averages across all three data-sets/indications (RGEP2) was considerably less accurate. When considering the distance from the bisection (as shown in Fig. 5) as a measure of estimation accuracy, we find that there is a consistent over-estimation of regulatory T cells. The estimation of regulatory T cells is confounded by the estimation of non-regulatory CD4+ T cells that are similar in their expression profile. There is a corresponding underestimation for the non-regulatory CD4+ T cells that is not as clearly visible because the overall percentage of non-regulatory CD4+ T cells is higher than the percentage of regulatory T cells. Despite this bias for these T cell subtypes, the estimation of the clinically relevant T cell ratios is unaffected.

To explore the impact of similar cell type profiles or missing cell type profiles on the estimation accuracy, we systematically evaluated cases where one cell type profile at a time was removed from RGEP3 (Supplementary Fig. 4). For most of the cases and cell types, the estimation accuracy is not affected by removing other cell type profiles. The estimation accuracy of CD4+ T cells, macrophages/monocytes, and malignant cell types were robust to all the changes. We observed reduced estimation accuracy for some of the more closely related cell types. CD8+ T cell estimation accuracy is affected by removing CD4+ T cells, and regulatory T cells estimation accuracy is affected by both removing CD8+ or CD4+ T cells. B cell accuracy is affected by

removing macrophages/monocytes. Dendritic cell accuracy is affected by removing B cells or macrophages/monocytes. NK cell accuracy is affected by removing CD8+ or CD4+ T cells. Endothelial cell and CAF accuracy is affected by removing melanoma cell profiles. To determine the impact of using alternative gene sets for the deconvolutions, we repeated the analyses using the best performing RGEP3 and four additional gene sets, as well as three alternative deconvolution algorithms. Interestingly, the impact of different gene sets and deconvolution algorithms was relatively small compared to the impact of the origin and quality of the RGEPs (see Supplementary Fig. 5). CIBERSORT in conjunction with the *Merged* gene set provided the best overall results.

**Validation of deconvolution results using independent data.** Using the RGEPs derived from single-cell RNA sequencing data, we established that the origin and quality of RGEPs impacts the accuracy of the deconvolution approach. Therefore, we propose that RGEPs derived from single-cell RNA sequencing data originating from the tissue of interest should be used for bulk deconvolution. The clinical data to which the deconvolution approach is applied, however, would be obtained by regular bulk RNA sequencing. Therefore, it is important to demonstrate that that RGEPs derived from single-cell RNA sequencing are appropriate for data measured by bulk RNA sequencing. To validate the deconvolution results on actual (rather than artificial) bulk data, we profiled three of the four ovarian cancer ascites samples additionally with bulk RNA sequencing and applied the deconvolution approach using RGEP3 to obtain estimates of the cellular composition of the samples. Further, using the same three samples, we quantified the cellular composition experimentally by FACS and by single-cell RNA sequencing followed by algorithmic

cell-type classification. Figure 6a shows a schematic of the data generation for these three samples, and Fig. 6b shows the quantitative comparison on the results obtained by the three different methods (see also Supplementary Tables 1 and 2 for details). Overall, the results are in good agreement. As all three methods have intrinsic biases, they only provide an estimate of the cellular composition of the samples. The biases are expected and can originate from differences in sample processing that can pose stress on the more fragile immune cells. In our validation data, we consistently observe a reduced estimate for the macrophage/ monocytes population when quantified by FACS. The single cell-based classification consistently estimates the highest proportion of macrophages/monocytes within this sample set. Similarly, the deconvolution approach consistently estimates a lower proportion of CD4+ T cells and similarly for the low abundant dendritic and NK cell populations.

**Estimation of tumour cell gene expression profiles**. Although using RGEP3 that is indication-specific but not patient-specific enables the accurate estimation of cellular composition of any given patient biopsy from bulk gene expression data, the gene expression profile of the malignant cells varies the most from patient to patient. Differences in gene expression in tumour cells are expected to play a key role in predicting response to traditional therapies, including both targeted and chemotherapies. As such, it is also of interest to estimate the patient-specific tumour cell profile following deconvolution. If consensus profiles exist for every non-malignant cell type and indication, the patient-specific tumour cell profile can be obtained by simply subtracting the profile of each non-malignant cell type from the bulk profile, weighted by its inferred proportion. In practice, however, the bulk profile will always be "contaminated" by cells for which consensus profiles do not exist ("unknown" cells). For example, neutrophils are not represented in scRNA-seq data, as they are difficult to isolate, highly labile ex vivo and therefore are difficult to preserve with current single-cell isolation methods[16]. Using scRNA-seq data, we calculated the estimated tumour cell expression profiles for each patient sample and compared them to the true tumour cell profile (Fig. 7a). As some genes, such as housekeeping genes, correlate between all cells irrespective of cell type, a certain baseline correlation is expected. We estimated this baseline correlation by correlating the gene expression profiles of the non-malignant cells with the true tumour cell gene expression profiles. We observe a baseline Pearson correlation $\rho$ of 0.7–0.8 for all samples, irrespective of the estimated proportion of tumour cells in the samples. As expected, the estimation accuracy of the tumour cell expression improved with increasing tumour cell content (Fig. 7b). Notably, when the estimated proportion of tumour cells in the samples exceeded 20%, the estimated tumour cell gene expression profiles exhibited a Pearson correlation of $\rho>0.9$ with the true profile. The predicted tumour cell gene expression profiles in samples with more than 20% but less than 70% tumour cells correlate better with the true tumour cell gene expression profiles compared to the uncorrected overall gene expression profiles. If a sample contains more than 70% tumour cells the gene expression profile of the whole sample is dominated by the tumour cells already and does not require any subtraction. For samples with less than 20% tumour cells, the subtraction does not improve the estimation because the signal of the tumour cell gene expression is low. In addition, the gene expression profile of the whole sample also does not provide information on the tumour cell profiles over the negative control, which is the non-tumour profiles in this case. In summary, for samples with a tumour cell content between 20 and 70% deconvolution results in significantly improved gene expression profiles.

## Discussion

Cellular heterogeneity is present in any biological sample. Single cell RNA-seq allows us to understand how cellular heterogeneity contributes to function or patient outcome. However, it is still much easier to obtain bulk gene expression data. The work presented here shows how deconvolution approaches can be applied to bulk gene expression data to infer cellular composition and to provide a tool to link cellular heterogeneity to biological function or drug response from bulk gene expression data. We show that with indication-type and cell type-specific RGEPs deconvolution methods like CIBERSORT can accurately estimate the cellular composition of a given biopsy sample and in addition give us more accurate information about the tumour cell gene expression profiles by eliminating contamination from non-malignant cells. This is most relevant if the tumour cell content ranges between 20 and 70%. Our work showcases the feasibility of deriving cleaner tumour cell gene expression profiles. The number of tumour samples in our study, however, is too limited to allow for representative biological conclusion on tumour cell biology in these patient samples.

Benchmarking different gene expression reference profiles and different deconvolution algorithms, we showed that the estimation accuracy is ultimately limited by the origin and quality of the RGEPs. RGEPs derived from PBMCs are insufficient to enable accurate deconvolution of tumour bulk gene expression data[17]. By combining well-established deconvolution algorithms with state-of-the-art single-cell RNA-seq data of tumour biopsies, we showed that indication-specific consensus profiles of immune, stromal, and tumour cells, obtained directly from the tumour microenvironment, can be used to obtain accurate estimates of the cellular composition of a given sample. Patient to patient variability, however, will continue to be a confounding factor for deconvolution methods. One strategy to address patient to patient variability is to analyse a large set of matched tumour and blood samples from different tumour types[18] and to quantify the impact of patient to patient variability on the proposed deconvolution method. Such data set of matched tumour and blood samples is necessary to enable more extensive cross-validation studies to prove that indication-specific consensus profiles provide accurate results across a larger population of patients. Overall, we found that the origin and quality of the reference profiles play a more dominant role than the deconvolution algorithm or gene set that is used, although gene sets designed to address as many cellular subsets as possible are clearly needed for accurate estimations of cellular heterogeneity.

With the availability of public scRNA-seq data from PBMCs and melanoma samples, as well as the ability to generate scRNA-seq data ourselves, we found that the gene expression profiles of tumour-associated immune cells differ considerably from those of blood-derived immune cells. Despite this systematic modulation, we found that patient-to-patient differences do not confound the deconvolution of bulk expression data and that consensus reference profiles can be established for each cell type, including tumour cells, for each specific microenvironment and indication.

We restricted our analyses to nine major cell types and three T cell subsets. Additional subdivisions can be added by defining these cell types in the scRNA-seq data-set and by choosing an appropriate gene set to enable these subdivisions. In practice, it is best to limit the number of subdivisions as much as possible, as more uncertainty is introduced when attempting to distinguish cell types with similar profiles. In our analysis, we see limitations of the approach for distinguishing between CD4+ (non-regulatory) T cell and regulatory T cells. As can be seen in Fig. 1, these two cell types form a continuum rather than distinct clusters. This is expected, as regulatory T cells are also CD4-positive and are a subset of CD4+ T cells in general. With larger scRNA-

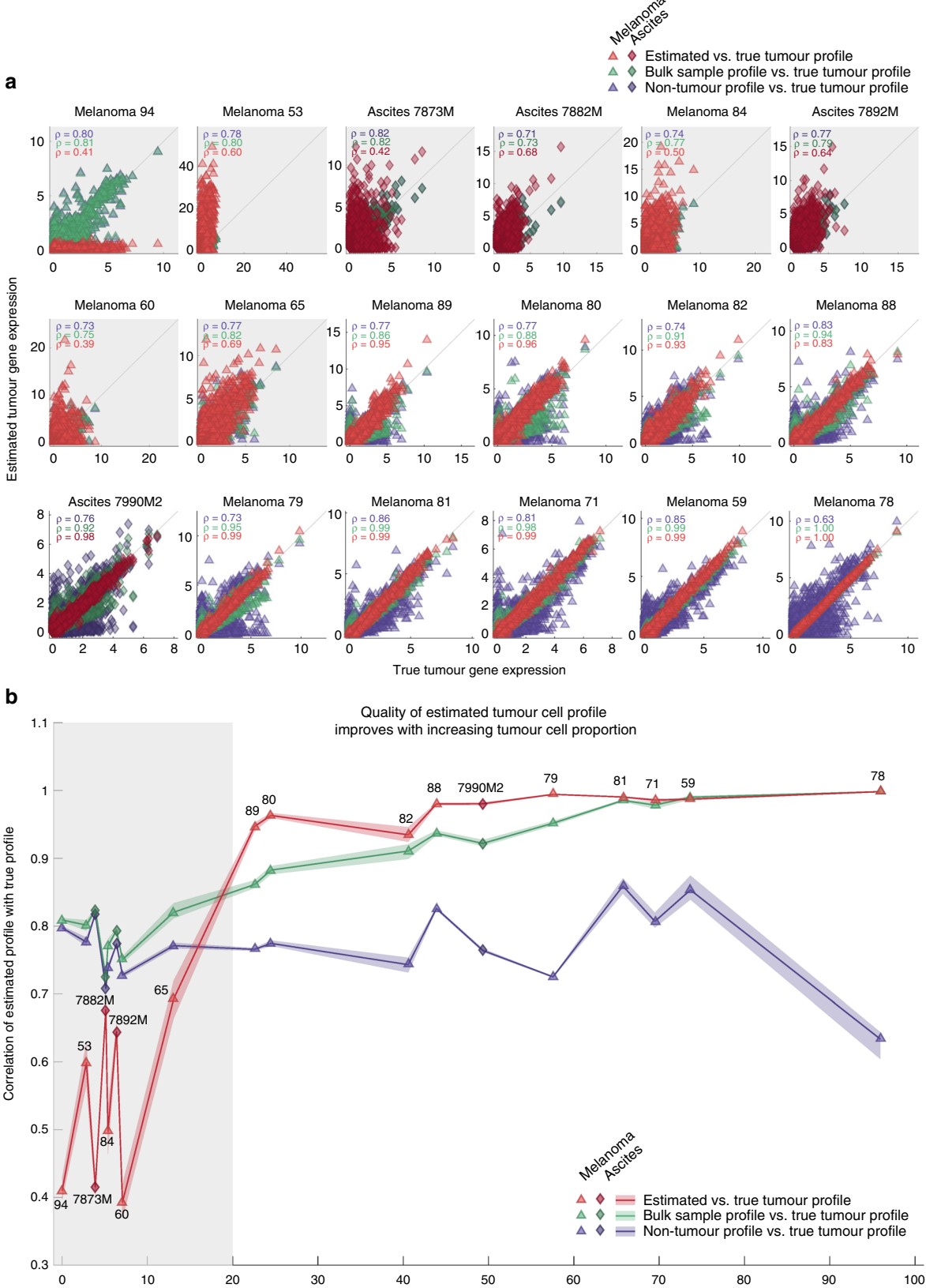

**Fig. 7** Estimation accuracy of patient-specific tumour cell gene expression profiles. **a** Scatter plot of estimated vs. true transcriptome wide gene expression (17,933 genes) of the tumour cells for individual patient samples. Patient samples without any tumour cells have been excluded from this analysis. $\rho$ denotes the Pearson's correlation. Correlation plots with grey background indicate patient samples with less than 20% tumour cell content. Colours according to legend in panel **b**. **b** Correlation values from panel **a** plotted against the estimated proportion of tumour cells for each patient sample. Shading represents uncertainty based on bootstrapping. Symbols and numbering denote individual patient samples

seq data-sets available in the future, refined RGEPs can be developed that might help to better distinguish between closely related cell types.

Given that we obtained the best results using indication-specific and cell-specific RGEPs, it is likely that consensus reference profiles for immune and stromal cells will need to be established from scRNA-seq data for every solid tumour indication. Why, then, are these deconvolution approaches necessary? At this time, scRNA-seq experiments are difficult to perform in a routine clinical setting. Cryopreservation protocols are currently being developed that will help to make scRNA-seq data more easily accessible[19]. It is not yet clear, however, how well these new protocols will work for tumour samples and the established protocols require the sample to be acquired and analysed within hours. Reference scRNA-seq data-sets from tumour samples can be obtained, but only in very controlled settings. With the appropriate data sets in place, however, deconvolution approaches enable routine clinical samples to be analysed both for cell content and patient-specific tumour cell gene expression profiles. Bulk RNA-seq data can easily be obtained from either flash-frozen or formalin-fixed, paraffin-embedded (FFPE) tissue samples, including both surgically resected material and core needle biopsies. The deconvolution approach presented here enables the estimation of immune cell content and improved tumour gene expression profiles in a clinical trial setting. The immune cell content of a tumour sample can also be determined by using more established multiplexed methods like immunohistochemistry (IHC) or immunofluorescence (IF)[20] or newer methods like imaging mass cytometry using FFPE tissue samples[21]. The advantage of these techniques is that a larger number of cells can be analysed and that these techniques also provide information about the spatial distribution of the different cell types.

However, these methods are limited to the number of proteins that can be analysed simultaneously currently (ranging from ~10 to 100), advantage of the deconvolution approach is that it is unbiased (i.e., hypothetical response markers do not need to be pre-specified). It allows one to link both the cellular characteristics and the cellular content with treatment response. We anticipate that this approach will aid in the discovery of novel predictive response biomarkers for both conventional and immune-directed therapy by taking cellular composition into account.

## Methods

**Ovarian cancer ascites samples**. Ovarian cancer ascites samples of four deidentified patients were obtained from The University of Massachusetts Cancer Center Tissue and Tumour Bank in Worcester, MA (https://www.umassmed.edu/ccoe/core-services/tissue-and-tumour-bank/). Samples were obtained from patients that provided their informed consented to the general procedural that remnant material can be used for research purposes (University of Massachusetts Medical School Institutional Review Board, approved protocol Docket #H-11731 with HIPAA waiver for discarded, de-identified specimens). Samples were shipped on ice on the same day and processed upon receiving. Each sample was filtered through a 70 μm filter and the cells were centrifuged down at ~300×g in a swing bucket centrifuge. Cells were re-suspended in PBS–5% FBS and counted. 1e7 viable cells were frozen down in 90% FBS + 10% DMSO and stored at −80 °C until use. Cryopreserved cells were thawed at 37 °C water bath. Cells were spun down and re-suspended in PBS-0.1% BSA and stored on ice. Viable cell number was measured, and cells were diluted to 1.6–2e5 cells per ml in PBS–0.1% BSA. About 3000 cells were encapsulated using the InDrops procedure[23,24] at the Single Cell Core at Harvard Medical School. The libraries from about 1000 cells per sample were sequenced with the Illumina Nextseq 500 method at the Molecular Biology Core Facility at Dana-Farber Cancer Institute. Total RNA was isolated from the same samples using RNeasy kit (Qiagen) and sent to Beckman Genomics for sequencing library construction and Illumina RNA sequencing.

**Data processing**. Gene expression values were used on the transcripts per million (TPM) scale as provided by current quantification methods[25–27]. The expression

values were transformed to

$$y = \log_2(\text{TPM} + 1).$$

To ensure cross-sample comparability, all single-cell melanoma, PBMC, and ascites samples were normalised to the average expression of 3559 housekeeping genes[28] by

$$\tilde{y}_i = y_i \cdot \frac{\overline{\text{HK}}}{\text{HK}_i},$$

where $y_i$ represents the gene expression profile of the $i$th sample, $\text{HK}_i$ denotes the average gene expression over all housekeeping genes of the $i$th sample and $\overline{\text{HK}}$ is the average expression over all housekeeping genes and samples. Other normalisation methods-like upper quartile or median normalisation could not be applied to scRNA-seq data as the single-cell measurements contain too many genes with zero expression leading to a zero-upper quartile and median for several samples. Gene symbols of the single-cell melanoma data were corrected to account for automatic conversion into dates by Microsoft Excel[29].

**Flow cytometry analysis**. Ascites were stained with either anti-human CD45 (H30, APC), CD3 (OKT3, Alexa-Fluor 700), CD4 (SK3, APC-Cy7), CD8 (RPA-T8, BV510), CD25 (BC96, Percp-Cy5.5), CD56 (5.1H11, BV570), CD127 (A019D5, BV421) CD16 (3G8, BV650) Abs or with anti-human CD45 (H30, APC), CD1c (L161, BV421), HLA-DR (L243, APC-Cy7), CD14 (M5E2, Alexa-Fluor 700), CD15 (W6D3, BV605) CD16 (3G8, BV650) Abs in PBS 2% FBS for 20 min at 4 °C. The antibodies were purchased from BD Pharmingen or Biolegend. The samples were acquired on an LSRFortessa flow cytometer (Becton Dickinson) and analysed using FlowJo software.

**Classification of cell types based in single-cell**. For classification of scRNA-seq data, a multi-step approach was developed. In contrast to the classification approach presented by Tirosh et al.[11], malignant and non-malignant cells were not treated separately, and only the scRNA-seq gene expression data were used for generating a training set. A workflow chart of our classification approach is depicted in Supplementary Fig. 6. After normalising all data as described above, t-SNE mapping was performed on the Merged gene set to identify clusters of similar cells. Subsequently, the DBSCAN algorithm[30] was used to identify clusters based on the t-SNE map as shown in Fig. 1a. The parameters of DBSCAN were set manually to MinPts = 25 and Eps = 1.5 with the aim that each larger cell group on the map is assigned to a separate cluster. For each cell, the expression of a total of 45 marker genes (see Supplementary Table 3) was normalised to [0, 1] and evaluated based on three categories of genes: (1) AND genes that are all required, (2) OR genes where only the expression of one of them is necessary, and (3) NOT genes where the expression is a negative selection criterion. Evaluating all three categories for each cell type led to a score describing the likelihood of each cell to belong to a certain cell type. The resulting score is depicted as a heat map on top of the t-SNE map in Supplementary Fig. 6b. In each DBSCAN cluster, a total cell type score was calculated and only cell types with a predominant total score (>75% of the maximal score) were assigned preventing the misclassification of closely related cell types (e.g. NK cells and T cells). This initial cell type assignment led to a sparse training set as depicted in Supplementary Fig. 6c. A decision tree classifier was trained based on these training data (also based on the Merged gene set). Using the trained classifier, the identity of all cells was predicted and validated based on five-fold cross-validation (Supplementary Fig. 6d) showing a high accuracy (98.06%) for the classification of the major cell types. Cells with a posterior probability lower than 0.99 were marked as "unknown". The resulting classification is shown in Supplementary Fig. 6e. Further, three sub-types of T cells (CD4+, CD8+, and regulatory T cells) were classified based on the T cell population defined in the first round of classification. The same procedure as explained above was repeated on the T cell subtypes (as shown in Supplementary Fig. 7). This was necessary because the similarity of sub-types is much higher than for distinct cell types. Only the parameters for DBSCAN needed to be adjusted (MinPts = 25 and Eps = 1.75) to account for the smaller sample size and the different distances on the t-SNE map of the T cells. The cross-validation of the classification resulted in an accuracy of 93.88% for the T cell sub types. The resulting t-SNE map indicating all cell types and T cell subtypes is depicted in Fig. 1a.

**Construction of artificial "bulk" gene expression data**. The artificial "bulk" gene expression data that was used for testing the deconvolution approach was generated from single-cell RNA sequencing data by aggregating reads from all cell barcodes for each patient sample. As single-cell and conventional bulk sequencing differ in their quantification biases, we cannot assume that single-cell-based RGEPs are applicable for deconvolution of conventional bulk sequencing data. Therefore, to apply the deconvolution based on single-cell RGEPs, conventional sequencing must be adapted to closely mimic the quantification process in single-cell sequencing, however, without the cell barcoding that would be problematic in a clinical trial setting.

**Deconvolution algorithms**. Computational approaches to decipher the relative immune cell content in the tumour environment from microarray or RNA-seq gene expression data have been proposed[8,9,31,32] and have been validated on blood samples[8,9] or in vitro cell mixtures[32]. Detailed reviews on this topic are available[7,33]. A method called CIBERSORT was proposed and its performance was compared to previously existing methods[8]. Using blood samples of a total of 41 patients, the authors could show that CIBERSORT outperforms the other methods when comparing the deconvolution results to flow cytometry data. The authors of the study also released a set of 547 genes (called LM22), which was used for their deconvolution approach. The CIBERSORT method (and most of the other methods mentioned above) assumes that the gene expression profile of an unknown bulk sample can be explained by the weighted sum of the cell type-specific profiles of which it is composed. The weights vector leading to the linear combination can be obtained by solving a linear equation system computationally. As biological data can be obscured by technical and biological variabilities, methods for deconvolution need to be robust against noise. The contamination of the sample with unknown cell types can be a further source of noise. A method called $v$-Support vector regression ($v$-SVR) combines feature-selection with a linear loss function and $L_2$-regularisation[34] and is therefore robust against noise[33]. The performance of deconvolution approaches has been widely demonstrated on in vitro mixtures and setting where the cellular gene expression profiles were directly measured, such as for peripheral blood mono-nuclear cell (PBMC) content in blood. Although there have been attempts to use RNA-seq data for deconvolution of cell mixtures[32,35], so far, the accuracy of the approach has not been evaluated in a realistic setting and in a systematic manner. The potential of having absolute expression values from RNA-seq data rather than relative data from microarray has not been fully exploited.

**Signature gene sets**. The basis for an accurate deconvolution is the choice of the *signature gene set*. The gene expression levels of these genes need to be informative enough to distinguish between cell types contained in the mixture/bulk sample. For our comparison, we chose five different signature gene sets. The LM22 gene set[8] consists of 547 genes of which 496 are contained in the scRNA-seq PBMC, melanoma, and ascites data-sets. The Table S12 gene set[11] contains 244 genes that are preferentially expressed in regulatory T cells of which 239 are present in all three data-sets. The Table S3 gene set[11], a list of 391 genes that have been identified as differentially expressed among the cell types in scRNA-seq data. Therefore, 374 genes are contained across all three scRNA-seq data-sets. A *Merged* gene set, generated by merging all genes from the LM22, the Table S3 and the Table S12 gene sets and adding the 45 marker genes used for classification training. It consists of 1076 unique genes, with 1015 genes in common with the scRNA-seq data. An *All genes* gene set, consisting of 17,936 genes that are contained in the all three scRNA-seq data-sets with 17,933 non-zero genes for at least one single-cell profile.

**Settings for algorithm comparison**. For deconvolution of the bulk patient profiles, the data was filtered to the Merged gene set and one of three deconvolution algorithms was applied. For $v$-SVR we used the implementation of libSVM[36] for MATLAB (version R2016a, The MathWorks Inc., Natick, MA, USA). The parameters were set to "$-s$ 4 $-t0$ $-n$ 0.50 $-h$ 0 $-c$ 1 $-q$". The mldivide function from MATLAB uses the pseudo-inverse of the matrix $B$ for solving for $w = \text{pinv}(B)^*m$. This is equivalent to using $w = \text{mldivide}(B, m)$. The fitlm function from MATLAB fits a linear model to the data based on a least-squares fit. The main difference to the mldivide function is that for fitlm an intercept is considered. For the CVX package for MATLAB[37] use used the setting lambda = 1 and solved using the SDPT3 algorithm[38] for semidefinite-quadratic-linear programming problems (see Supplementary Note 1).

**Processing of estimation results**. The results for the proportions of known cell types $\vec{w}$ as obtained by one of the above-mentioned algorithms are processed by replacing negative numbers by zeros[8]. The proportion of unknown other cell types $\tilde{w}$, i.e. cell types for which no reference profile was available, is calculated by taking the difference between one and the sum of all $m$ known cell proportions:

$$\tilde{w}_i = 1 - \sum_{i=1}^{m} w_i.$$

**Estimation quality assessment**. To assess the quality of our deconvolution results, we compared the true cellular fractions, as calculated from the number of single-cell measurements for each cell type and patient, with the estimation result by calculating Pearson's correlation coefficient $\rho$ for all patients. We quantified the uncertainty of our quality measure by performing bootstrap re-sampling (100 replications) of our deconvolution results and calculated the median and lower and upper quartiles.

**Dimensionality reduction**. To obtain a low-dimensional representation of high-dimensional data, dimensionality reduction methods can be applied. t-SNE enables the reduction from many to two dimensions while keeping local neighbourhoods[13]. For removing noise and improving the performance, a principle component analysis can be used to reduce the initial dimensionality before running t-SNE. We used the Barnes–Hut implementation of t-SNE[39] with the default settings to analyse our data. The result is a map that reflects the similarities between the high-dimensional input data as depicted in Fig. 1a and Supplementary Figs. 5 and 6.

**Tumour gene expression profile estimation**. To calculate the gene expression profile of an average tumour cell for each individual patient, we need to subtract the explained portion of gene expression from the patient's bulk sample gene expression profile and rescale the expression with the estimated tumour proportion, i.e.

$$\vec{t}_i = \frac{\vec{m}_i - \underline{B}_{\text{non-tumor}} \vec{w}_{i,\text{non-tumour}}}{w_{i,-\text{tumour}}}$$

**Code availability**. Source code for running the benchmark study and producing the figures is available at https://figshare.com/s/711d3fb2bd3288c8483a. The corresponding data-sets can be obtained using the download scripts provided in the source code repository.

**Data availability statement**. Single-cell melanoma data were obtained from Gene Expression Omnibus (GEO; https://www.ncbi.nlm.nih.gov/gds) under accession number GSE72056 in a pre-processed format. Single-cell RNA-seq data of PBMCs from patient blood samples were downloaded from the 10x Genomics website (https://support.10xgenomics.com/single-cell/datasets; "4k/8k PBMCs from a Healthy Donor", "Frozen PBMCs (Donor A/B/C)"[12]) and 1000 random cells were selected randomly for each donor to ensure similar size as for the melanoma and ascites data-sets. The ovarian cancer ascites data has been deposited together with the source code and is available at https://figshare.com/s/711d3fb2bd3288c8483a. The raw Ovarian Cancer ascites RNA-seq read data has not been submitted to a public repository due to privacy concerns, but can be made available upon reasonable request to the corresponding author pending appropriate approval from study participants. The authors declare that all the other data supporting the findings of this study are available within the article and its supplementary information files and from the corresponding author upon reasonable request.

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

## Acknowledgements

We thank the lab of Allon Klein at Harvard Medical School for providing support to pre-process single-cell RNA sequencing data from the InDrops platform and the internal review team at Merrimack Pharmaceuticals, Inc. for reviewing the article prior to submission. We also thank Merrimack Pharmaceuticals, Inc. for sponsoring the research and supporting publication of the results. Further, we would like to acknowledge The University of Massachusetts Cancer Center Tissue and Tumour Bank for providing ovarian cancer ascites samples.

## Author contributions

All co-authors helped to write and to review the manuscript. M.S. preformed all computational analysis. S.F. processed the ovarian cancer ascites samples and preformed the flow cytometry analysis. J.D. processed the ovarian cancer ascites samples and generated the single-cell and bulk data. N.R. helped to pre-process and analyse the ovarian cancer ascites data. E.K., G.M. and B.S. helped to design and support the study. A.R. designed and executed the study.

## Additional information

**Competing interests:** The authors and Merrimack Pharmaceuticals, Inc. declare no competing financial interest.

