## [Peer Review File · Nature Communications]

Reviewer #1:
(Remarks to the Author):

General comments

The manuscript by Schelker et al. describes a deconvolution-based method for bulk gene expression data using reference gene expression profiles from tumor-derived single-cell RNA sequencing data. The authors used two published single-cell RNA-seq data sets (one data set from tumors from 19 melanoma patients and one data set from PBMCs from healthy donors) and generated one data set from 4 ovarian cancer ascites.

There is an urgent need to develop and validate deconvolution methods in order to characterize cell populations in complex samples. While this study is an important step towards this goal, there are serious shortcomings. First, the authors compare expression profiles from tumor-associated immune cells from patients and PBMCs from healthy volunteers. There are both, technical and biological issues with this approach. As the authors correctly stated, comparison of single-cell RNA-seq data from different sources and technologies is difficult. Hence, it is necessary to provide additional information and data that the approach is valid. But more important is the biological: it is necessary to compare tumor-infiltrating immune cells and PBMCs from the same patient (see for example paper from De Simone et al, Immunity 2016). Second, the authors used CIBERSORT deconvolution. However, CIBERSORT was developed using microarray data and the method has to be scaled up for RNA-sequencing data. And third and most importantly, independent validation using either IHC/IF or FACS data is missing.

Specific comments

1. In Figure 1 it is difficult to distinguish between PBMCs, melanoma and ascites.
2. It is unclear how the T cell subsets in Figure 3 were calculated. The
3. What was the reason for not using the classification approach from Tirosh et al (i.e. treating separately malignant and non-malignant cells)?
4. How representative are cells from ascites cell for tumor cells?
5. What is the performance of the deconvolution algorithm compared to other published algorithms (e.g. Li et al. Genome Biology 2016) ?
6. References 12 and 13 are the same.
7. Typos should be corrected (e.g. fresh-frozen)

Reviewer #2 :
(Remarks to the Author):

The authors study the impact of reference gene expression profiles (RGEPs) on the performance of cell type composition deconvolution approaches. The authors derive RGEPs from both publically available and in-house single-cell RNAseq data for PBMCs and patient samples for various cancers.

The newly derived RGEPs for cancer tissues constitute the original contribution of this work. This study demonstrates the crucial importance of considering the right RGEPs to make bulk sequencing deconvolution approaches work. However, I have a few concerns regarding the practical implications and significance of these results. See below for details.

Major concerns:

It seems that deconvolution performance is assessed on synthetic bulk data, constructed by in silico pooling of real single-cell transcriptomic data. Since such deconvolution approaches would

practically be applied to real bulk transcriptomic data, it would be instructive and convincing to demonstrate deconvolution on such real data. I would expect that deconvolution performance would suffer from divergence of measurement noise distributions in bulk measurements wrt single-cell measurements.

The deconvolution approach assumes that all reference cell types are known and represented with a reference profile. How much is the analysis compromised by cell types present in the sample, but not included as reference? It seems that in this situation actually low frequent cell types, such as dendritic cells, can have highly inflated cell type proportions (Fig. 2a, results for RGEP1).

How well can RGEPs be generalized to new samples? What would be the deconvolution accuracy for RGEPs derived from a set of training patients, but evaluated on a set of unseen test patients?

What is the deconvolution accuracy of low abundant cell types? It would be interesting to see a plot that shows deconvolution accuracy depending on true cell type proportion. Is there a consistent trend towards lower accuracy with decreasing proportion and to if so what is the magnitude of this trend.

Fig. 3: Why is there a consistent overestimation of regulatory T cell proportion? Performance evaluation by correlation masks this phenomenon. Why not report deviation from identity, e.g. by square loss?

Fig. 3: which "constructed" bulk data was the basis for the analysis of T cell subsets? The large proportions for the T cell subtypes suggest that the considered "constructed" bulk data has been generated from T cell single cell data alone? If so, what would be the respective results for the "constructed" bulk data from all single cell data, comprising all other cell types?

Fig 4: It seems that the reconstructed tumor profile is significantly worse to the estimate with the bulk profile estimate, where most interesting, i.e. lower fraction of tumor cells. These results indicate that deconvolution is not suited to reconstruct patient specific cell type profiles. This limitation should be discussed in the Results/Discussion to indicate the scope of deconvolution methods, limited to estimate cell type fractions for a priori defined cell types.

What kind of questions can be answered deconvolution approaches, that could not be addressed by single-cell sequencing experiments? The authors allude to applications in immunotherapy, where cell type proportions/ratios are associated with clinical covariates. The practical value of the proposed approach would become apparent if the authors demonstrated on suitable examples.

The deconvolution approach assumes singular reference profiles for cell types. Transcript profiles happen to follow a, possibly cell type specific distribution. This notion of a cell type further precludes the situation of a class of related and dispersed class of cell types as observed for myeloid or T helper cells. How sensitive are the considered published deconvolution approaches for this phenomenon? How can this be accounted for in the deconvolution, if relevant?

Minor concerns:

Title misleading "Estimation of immune cell content in tumour tissue using single-cell RNA-seq data". Title suggests that immune cell content is estimated from single-cell RNA-seq data. However, the point of this work is such estimation from bulk-measurements.

Reviewer #1:

General comments

The manuscript by Schelker et al. describes a deconvolution-based method for bulk gene expression data using reference gene expression profiles from tumor-derived single-cell RNA sequencing data. The authors used two published single-cell RNA-seq data sets (one data set from tumors from 19 melanoma patients and one data set from PBMCs from healthy donors) and generated one data set from 4 ovarian cancer ascites. There is an urgent need to develop and validate deconvolution methods in order to characterize cell populations in complex samples. While this study is an important step towards this goal, there are serious shortcomings.

We thank the reviewer for his/her encouraging words.

First, the authors compare expression profiles from tumor-associated immune cells from patients and PBMCs from healthy volunteers. There are both, technical and biological issues with this approach. As the authors correctly stated, comparison of single-cell RNA-seq data from different sources and technologies is difficult. Hence, it is necessary to provide additional information and data that the approach is valid.

We have provided additional validation data that demonstrates that the deconvolution approach remains valid on data from different sequencing technologies (single-cell vs. bulk sequencing) in a new section titled "Validation of deconvolution results using independent bulk sequencing data set on ascites samples". Using the new data, we could demonstrate that the deconvolution approach that was trained on single-cell data performs as desired on independent data from bulk RNA sequencing (see new Figure 4).

But more important is the biological: it is necessary to compare tumor-infiltrating immune cells and PBMCs from the same patient (see for example paper from De Simone et al, Immunity 2016).

We agree with the reviewer that this is a very important question. The main message of our manuscript is to raise awareness of the importance of the tissue context and how it impacts the results from the deconvolution approach. Currently, no public single-cell RNA sequencing data sets from matched tumor and blood samples are available. We recognize that we might not have stated the main message of our manuscript clearly enough. To improve the discussion of this important topic, we rewrote parts of the introduction and discussion sections and discussed the recent publication by De Simone et al., as suggested by the reviewer.

Second, the authors used CIBEROSRT deconvolution. However, CIBERSORT was developed using microarray data and the method has to be scaled up for RNA-sequencing data.

The reviewer is correct, CIBERSORT has been established originally for microarray data. One contribution of our work is that we have adapted the algorithm with the Support Vector Regression being its main methodological component to RNA-sequencing data. To this end, we had to develop new reference

gene expression profiles that are based on RNA-seq data. We have already proven with the results displayed in the manuscript that the adapted approach has desired properties and, given the appropriate reference gene expression profiles, the underlying true composition of the cell mixture can be inferred accurately (Figures 2, 3 and 4). During this process, however, we discovered the discrepancy and performance loss caused by reference gene expression profiles that were derived from an inappropriate biological context (e.g. PBMC vs. tumor microenvironment). We realize that this was not sufficiently described and worked on clarifying this point by in the section “Origin and quality of RGEP determine deconvolution results”.

And third and most importantly, independent validation using either IHC/IF or FACS data is missing.

We thank the reviewer for this valid comment. We have obtained independent bulk RNA-sequencing data, ran the deconvolution and compared the results to an independent FACS-based quantification. The additional validation data is presented in the new section titled “Validation of deconvolution results using independent bulk sequencing data set on ascites samples”. The new data shows that the deconvolution results and the FACS-based quantification are in good agreement.

Specific comments

1. In Figure 1 it is difficult to distinguish between PBMCs, melanoma and ascites.

We agree with the reviewer and improved the graphical display by adding the new Extended Data Fig. 1 that shows the distinction between PBMCs, melanoma and ascites more clearly.

2. It is unclear how the T cell subsets in Figure 3 were calculated. The

We improved the description of how the T cell subsets were calculated in the section “Origin and quality of RGEP determine deconvolution results”. (It seems that the comment of the reviewer is incomplete?)

3. What was the reason for not using the classification approach from Tirosh et al (i.e. treating separately malignant and non-malignant cells)?

The classification in Tirosh et al. is a manual approach. We aimed to create an algorithm that can be applied more easily on additional data in the future. In addition, we had difficulties to reproduce the malignancy score used in Tirosh et al.

4. How representative are cells from ascites cell for tumor cells?

This is a difficult question, and we don't know if we can answer it sufficiently. As the reviewer has noticed, the percentage of tumor cells in the ovarian cancer ascites samples is relatively low in 3 of the 4 samples. We do not think that biological statements about the phenotype of these ovarian cancer cells would be justified. However, we think that using these four samples as test cases for the deconvolution approach, together with the sample from the other sources, is justified. We discuss this in the Discussion part of the manuscript now.

5. *What is the performance of the deconvolution algorithm compared to other published algorithms (e.g. Li et al. Genome Biology 2016)?*

The key message of our manuscript is to raise awareness that the results of deconvolution approaches are generally dependent on the RGEPs that are used and that the RGEPs should be derived from the respective tissue that is studied. We compared several methods and find that these results are independent of the method or gene set that is used (see Extended Data Figure 5). Our manuscript does not aim at an in-depth methods comparison. We cite an analysis by Newman et al. Nature Methods that has extensively compared several methods and found that CIBERSORT is the best performing method.

6. *References 12 and 13 are the same.*

We apologize for this mistake, it is corrected now.

7. *Typos should be corrected (e.g. flesh-frozen)*

We apologize and hope that all typos are corrected now.

Reviewer #2 :

The authors study the impact of reference gene expression profiles (RGEPs) on the performance of cell type composition deconvolution approaches. The authors derive RGEPs from both publically available and in-house single-cell RNAseq data for PBMCs and patient samples for various cancers.

The newly derived RGEPs for cancer tissues constitute the original contribution of this work. This study demonstrates the crucial importance of considering the right RGEPs to make bulk sequencing deconvolution approaches work. However, I have a few concerns regarding the practical implications and significance of these results. See below for details.

Major concerns:

It seems that deconvolution performance is assessed on synthetic bulk data, constructed by in silico pooling of real single-cell transcriptomic data. Since such deconvolution approaches would practically be applied to real bulk transcriptomic data, it would be instructive and convincing to demonstrate deconvolution on such real data. I would expect that deconvolution performance would suffer from divergence of measurement noise distributions in bulk measurements wrt single-cell measurements.

We thank the reviewer for this valid comment. We have provided additional validation data that demonstrates that the deconvolution approach remains valid on data from difference technologies in a new section titled "Validation of deconvolution results using independent bulk sequencing data set on ascites samples". Using new data, we could demonstrate that the deconvolution approach that was trained on single-cell data performs as desired on data from bulk RNA sequencing (see new Figure 4).

The deconvolution approach assumes that all reference cell types are known and represented with a reference profile. How much is the analysis compromised by cell types present in the sample, but not included as reference? It seems that in this situation actually low frequent cell types, such as dendritic cells, can have highly inflated cell type proportions (Fig. 2a, results for RGEP1).

The reviewer raises a very interesting and valid question here. To study the impact of potentially “missing” cell types on the deconvolution accuracy, we generated a set of systematic test cases. For each test case, we removed the reference gene expression profiles for one of the cell types. The results are shown in the new Extended Data Figure 5. For most of the cases and cell types, the estimation accuracy is not affected by removing other cell type profiles. The estimation accuracy of CD4+ T cell, Macrophages/Monocytes, and of the malignant cell types were robust to all the changes. We observed reduced estimation accuracy for some of the more closely related cell types. CD8+ T cell estimation accuracy is affected by removing CD4+ T cells and regulatory T cells estimation accuracy is affected by both removing CD8+ or CD4+ T cells. B cell accuracy is affected by removing Macrophages/Monocytes. Dendritic cell accuracy is affected by removing B cells or Macrophages/Monocytes. NK cell accuracy is affected by removing CD8+ or CD4+ T cells. Endothelial cell and CAF accuracy is affected by removing Melanoma cell profiles. We thank the reviewer for the valuable suggestion. The added data helps to clarify how deconvolution results are affected by missing cell types.

How well can RGEPs be generalized to new samples? What would be the deconvolution accuracy for RGEPs derived from a set of training patients, but evaluated on a set of unseen test patients?

These are difficult questions, and we don’t know if we can answer them sufficiently yet. Our results show that the RGEPs should be established (at least) for each major indication or tissue context. Patient to patient variability seems manageable based on the data we have currently. With our manuscript, we would like to raise awareness of this problem. Unfortunately, the sample size is currently too limited to allow for a cross-validation analysis. To stress this point more, we broadened the discussion on this important question and the need of further research on this. We hope that the message of our manuscript is now clearer.

What is the deconvolution accuracy of low abundant cell types? It would be interesting to see a plot that shows deconvolution accuracy depending on true cell type proportion. Is there a consistent trend towards lower accuracy with decreasing proportion and to if so what is the magnitude of this trend?

This is an interesting suggestion. Our results shown in Figure 2 demonstrate this point already. Focusing on the most relevant setting, RGEP3, we find that the accuracy (for this purpose measured as distance from the bisection) is relatively independent of the true cell type proportion. However, the overall accuracy is different for each cell type. We discuss this observation now in the section “Origin and quality of RGEP determine deconvolution results”.

Fig. 3: Why is there a consistent overestimation of regulatory T cell proportion? Performance evaluation by correlation masks this phenomenon. Why not report deviation from identity, e.g. by square loss?

The reviewer is correct. There is a bias for the estimation of the regulatory T cell subpopulation. The estimation is confounded in this case by the non-regulatory CD4+ T cells that have a similar gene expression profile with an almost continuous transition between the two subpopulations. There is a corresponding underestimation for the non-regulatory CD4+ T cells. The underestimation bias is not as clearly visible because the overall percentage of non-regulatory CD4+ T cells is higher than the percentage of regulatory T cells. Despite the bias for regulatory T cells, the clinically meaningful ratios of Tregs to other T cells is relatively unaffected. We improved the discussion of this observation in the section “Origin and quality of RGEP determine deconvolution results” and hope that this is clearer now.

Fig. 3: which “constructed” bulk data was the basis for the analysis of T cell subsets? The large proportions for the T cell subtypes suggest that the considered “constructed” bulk data has been generated from T cell single cell data alone? If so, what would be the respective results for the “constructed” bulk data from all single cell data, comprising all other cell types?

The reviewer is correct, the proportions in Figure 3 are referring to the T cell subset within the T cell population. The total proportion of T cells compared to the rest of the sample is displayed in Figure 2, together with the deconvolution results for T cells in total (not split into subtypes). We improved the description of Figure 3 to make this clearer.

Fig 4: It seems that the reconstructed tumor profile is significantly worse to the estimate with the bulk profile estimate, where most interesting, i.e. lower fraction of tumor cells. These results indicate that deconvolution is not suited to reconstruct patient specific cell type profiles. This limitation should be discussed in the Results/Discussion to indicate the scope of deconvolution methods, limited to estimate cell type fractions for a priori defined cell types.

We do agree with the reviewer that the reconstructed tumor profile is worse than the bulk profile estimate for samples that contain less than 20% tumor cells. For the samples that contain more than 20% tumor cells, however, the reconstructed tumor profile is significantly improved. We acknowledge that there is a limit on the applicability of this reconstruction of the tumor profile. Our results show as well that the bulk profile does not contain information on the tumor cells for samples with less than 20% tumor cells either. This can be seen by the “non-tumor profile vs. true tumor profile” (blue line) being overlaid with “bulk sample profile vs. true tumor profile” (green line) for samples with less than 20% tumor cells. We realize that we have not explained this sufficiently clear and improved the discussion of these results in the section “Estimation of patient-specific tumor cell gene expression profiles”.

What kind of questions can be answered deconvolution approaches, that could not be addressed by single-cell sequencing experiments? The authors allude to applications in immunotherapy, where cell type proportions/ratios are associated with clinical covariates. The practical value of the proposed approach would become apparent if the authors demonstrated on suitable examples.

We agree with the reviewer, single-cell sequencing could in principle make deconvolution approaches unnecessary. At this point, the single-cell sequencing technology is however far away from being applicable in routine clinical practice. We discuss this point in the Discussion section. The value of the

deconvolution approach therefore lies in its practicality and clinical applicability. We understand that this is less interesting from an academic standpoint, but could be very valuable for clinical studies. We hope that our argumentation is now clearer in the discussion.

The deconvolution approach assumes singular reference profiles for cell types. Transcript profiles happen to follow a, possibly cell type specific distribution. This notion of a cell type further precludes the situation of a class of related and dispersed class of cell types as observed for myeloid or T helper cells. How sensitive are the considered published deconvolution approaches for this phenomenon? How can this be accounted for in the deconvolution, if relevant?

This an important point and we thank the reviewer to bring this up. As the reviewer has already suggested, we also think that there is a limit to the accuracy of the deconvolution approach. Especially when cell (sub)types form a continuum, rather than clearly distinct clusters. For the major cell types in our study this is not a limitation, see Figure 2. As discussed in the above comment on the bias for regulatory T cells, we start to see this limitation for the distinction between CD4+ non-regulatory T cells and regulatory T cells. It is important to stress these limitations. As stated above, we improved the discussion of our results and hope that the limitations are clearer now.

Minor concerns:

Title misleading “Estimation of immune cell content in tumour tissue using single-cell RNA-seq data”. Title suggests that immune cell content is estimated from single-cell RNA-seq data. However, the point of this work is such estimation from bulk-measurements.

We thank the reviewer for this comment and agree. We would like to consider a title that makes this point more clear. However, the character length limitation for the title makes this very difficult. Therefore, we would like to get the feedback from the editor on this point before making any changes.

Reviewer #1 (Remarks to the Author):

General comments

The revised manuscript by Schelker et al. addressed all issues raised in the original review. Most importantly, the authors carried out additional validation experiments by profiling 3 of the 4 samples with bulk RNA-seq and additional FACS analysis. However, the results of the validation are not presented in an acceptable form and it is difficult to judge if the validation was successful. It is stated that the validation results are in "good agreement" but that is not clear from Figure 4. The results need to be provided additionally in a table so that also other cell type besides macrophages and CD4 cells are clearly visible.

Specific comments

1. The reference 18 (line 343) is not appropriate since De Simone et al did not use PBMCs and tumor samples to define a baseline for the deconvolution.

2. The last paragraph in the discussion is not convincing. The authors did not discuss alternative techniques like multiplexed IHC or IF (e.g. see recent paper by Tsujikawa et al., Cell Reports 2017 or commercially available systems and assays like from Vectra from Perkin Elmer). Multiplexed IHC/IF is better suited for clinical trials since it provides: 1) quantitative information of different cell types. The number of different cell types is similar or even larger than the one presented in the paper. 2) is based on FFPE samples which are easier to obtain. 3) provides also spatial information which is crucial in the context of cancer immunology (e.g. center of the tumor or invasive margin). The paragraph needs to be modified.

Reviewer #2 (Remarks to the Author):

see uploaded commented rebuttal letter. New reviewer comments are in red font.

Reviewer #1:

General comments

The revised manuscript by Schelker et al. addressed all issues raised in the original review. Most importantly, the authors carried out additional validation experiments by profiling 3 of the 4 samples with bulk RNA-seq and additional FACS analysis. However, the results of the validation are not presented in an acceptable form and it is difficult to judge if the validation was successful. It is stated that the validation results are in “good agreement” but that is not clear from Figure 4. The results need to be provided additionally in a table so that also other cell type besides macrophages and CD4 cells are clearly visible.

We thank the reviewer for his/her encouraging words and the good suggestions. We provided the validation data as additional table so the results are visible more clearly and quantitatively, see new Supplementary Table 2 and 3.

Specific comments

1. The reference 18 (line 343) is not appropriate since De Simone et al did not use PBMCs and tumor samples to define a baseline for the deconvolution.

We agree, our formulation in this line were not accurate. We modified the paragraph to “One strategy to address patient to patient variability is to analyse a large set of matched tumour and blood samples from different tumour types¹⁸ and to quantify the impact of patient to patient variability on the proposed deconvolution method. Such data set of matched tumour and blood samples is necessary to enable more extensive cross-validation studies to prove that indication-specific consensus profiles provide accurate results across a larger population of patients.” We hope the reviewer will find this more accurate.

2. The last paragraph in the discussion is not convincing. The authors did not discuss alternative techniques like multiplexed IHC or IF (e.g. see recent paper by Tsujikawa et al., Cell Reports 2017 or commercially available systems and assays like from Vectra from Perkin Elmer). Multiplexed IHC/IF is better suited for clinical trials since it provides: 1) quantitative information of different cell types. The number of different cell types is similar or even larger than the one presented in the paper. 2) is based on FFPE samples which are easier to obtain. 3) provides also spatial information which is crucial in the context of cancer immunology (e.g. center of the tumor or invasive margin). The paragraph needs to be modified.

This is a valid point and is now discussed in greater detail in the last part of the discussion.

Reviewer #2:

The additional validation data on real bulk data is interesting. The comparison of the different cell type frequency assignments in Fig. 4 seem to be dominated in magnitude by the abundant

Macrophage/Monocyte population. Why not report relative deviations from one of the frequency assignment methods, arbitrarily defined as reference? Relative deviations would be the more informative performance measure and not affected by absolute frequency of each cell type.

We thank the reviewer for this good suggestion, that is also consistent with the request of reviewer 1. We provided the validation data as additional table so the results are visible more clearly and quantitatively, see new Supplementary Table 2 and 3.

This systematic analysis of leaving out reference gene expression profiles for one of the cell types is instructive. To further evaluate the hypothesis that deconvolution accuracy is mostly affected by removal of related cell types it would be helpful to scatter plot deconvolution accuracy correlation for a specific cell type vs the correlation of the reference profile for this cell type and the removed cell type. If the hypothesis is correct, I would expect this plot to exhibit a strong dependency structure.

This is an interesting suggestion, we generated to scatter plot as proposed, see figure below. Indeed, there is a trend of reduced deconvolution quality (y-axis) with the scenarios where similar expression profiles are left out (x-axis, to the right). The trend is visible for example for: Regulatory T cells when the other T cell profiles are removed (blue dots); Dendritic cell when B cell or Macrophage profiles are removed (yellow and dark green dots); NK cells when CD8+ T cells are removed (medium blue dot); CAFs when melanoma or ovarian cancer cells are removed (pink and red dots). While these observations are interesting and make sense, we did not feel they were strong enough to include into the manuscript.

The complemented discussion helps to appreciate the masking or the joint over-/underestimation phenomena. This discussion would benefit by adding quantitative measures, such as square loss to the correlation measures in the plots of Fig. 3 and referring to these in the main text.

This is a valid comment. During our studies, we debated using MSE and other measures than correlation. However, we do believe correlation might be the most meaningful measure. As seen for regulatory T cells, there can be systematic biases (as discussed in the manuscript already) for some cell types. Using an *in-silico* analysis as we presented here, we can use the results to “calibrate” quantification for those cell types. Therefore, MSE and related measures would not capture this possibility. Therefore, we prefer to keep correlation as quantitative measure of accuracy.

Reviewer suggestion: “Single-cell RNA-seq data informs deconvolution of immune cell content in bulk measurements of tumor tissue”

We would be happy to consider this title and agree, it fit our manuscript much better. However, it is not compatible with the character length limitation of the journal (90 characters with whitespaces). Therefore, we would like to get feedback from the editor if this title could be used for our manuscript or not.

Reviewer #1 (Remarks to the Author):

The authors addressed satisfactorily all issues raised in the previous reviews. The paper is now acceptable for publication.

Reviewer #2 (Remarks to the Author):

The authors have addressed all main concerns. See below the remaining minor concern.

The complemented discussion and rebuttal comments help to appreciate the masking or the joint over-/underestimation phenomena. While the authors favor reporting correlation as quantitative performance measure in the plots of Fig. 3, it will also be instructive to see other commonly used measures such as square loss not masking differences in absolute levels. These additional measures do not necessarily have to be shown in the figure of the main manuscript, to avoid unnecessary cluttering, but could be shown separately in the Supplement.